# Placebo and nocebo in clinical practice: An online cross-sectional survey of healthcare professionals from European countries on views, practices and training needs

Mary O'Keeffe[1,2], Nathan Skidmore[2,3*], Arianna Bagnis[4*], Przemysław Bąbel[5], Elżbieta A. Bajcar[5], Alessandra De Palma[6], Andrea W.M. Evers[7,8,9], Eveliina Glogan[10], Julia W. Haas[10,11], Stefanie H. Meeuwis[7,8,12], Marek Oleszczyk[13], Antonio Portolés[14], Johan W.S. Vlaeyen[10,15], Katia Mattarozzi[4], on behalf of PANACEA Consortium

1 UCD School of Public Health, Physiotherapy and Sports Science, University College Dublin, Dublin, Ireland, 2 European Pain Federation EFIC, Brussels, Belgium, 3 Department of Sport and Exercise Science, Faculty of Social Sciences and Health, Durham University, Durham, United Kingdom, 4 Department of Medical and Surgical Sciences, University of Bologna, Bologna, Italy, 5 Pain Research Group, Institute of Psychology, Jagiellonian University, Kraków, Poland, 6 IRCCS Azienda Ospedaliero-Universitaria di Bologna, Policlinico di Sant'Orsola, Bologna, Italy, 7 Institute of Psychology, Leiden University, Leiden, the Netherlands, 8 Center for Interdisciplinary Placebo Studies (IPS) Leiden, Leiden, the Netherlands, 9 Medical Delta Healthy Society, Leiden University, Technical University Delft and Erasmus University Rotterdam, Rotterdam, the Netherlands, 10 Faculty of Psychology and Educational Sciences, KU Leuven, Leuven, Belgium, 11 Department of Psychology, University of Kaiserslautern-Landau (RPTU), Landau, Germany, 12 Department of Medical and Clinical Psychology, Tilburg University, Tilburg, the Netherlands, 13 Department of Family Medicine, Jagiellonian University Medical College, Krakow, Poland, 14 Department of Farmacología y Toxicología, Universidad Complutense Madrid, Madrid, Spain, 15 Experimental Health Psychology, Maastricht University, Maastricht, the Netherlands

¶ The complete membership of the author group can be found in the Acknowledgments.
* arianna.bagnis@unibo.it (AB); nathan.skidmore@efic.org (NS)

## Abstract

### Background

Placebo and nocebo effects significantly influence health outcomes, yet healthcare professionals receive limited training and guidance on their mechanisms and clinical application, creating a gap in education and practical understanding. Conducted within the European PANACEA Consortium, this study evaluated healthcare professionals' knowledge, attitudes, and practices regarding placebo and nocebo effects, and assessed their needs in further education.

### Methods

An online cross-sectional survey among a European multi-country convenience sample of healthcare professionals collected data assessing participants' knowledge, perceptions, and experiences regarding placebo and nocebo effects; their application and ethical considerations in clinical practice; and investigated educational needs

**Data availability statement:** The data underlying the results presented in this study cannot be shared publicly due to ethical restrictions and participant confidentiality. Data are available from the Bioethical Committee of the University of Bologna (contact via bioethics@unibo.it) for researchers who meet the criteria for access to confidential data. Researchers must provide a formal request and evidence of ethical approval for secondary data analysis. The survey instrument, CHERRIES checklist, and selected supporting qualitative quotes are available in the Supplementary Information files.

**Funding:** The study is funded by a grant from the Erasmus+ Program of the European Union (grant number: IT02-KA220-HED- 000088065).

**Competing interests:** M. O'Keeffe is supported by an Irish fellowship (UCD Ad Astra Fellowship).

and interest in further training. Quantitative data were analyzed using descriptive statistics, and thematic analysis was applied to the free-text responses.

## Results

Amongst 807 participants, 71.7% reported taking advantage of placebo effects in their practice, and over half of participants (55.8%) observing nocebo effects. Participants reported feeling somewhat confident (53.3%) in harnessing placebo effects with 47.5% feeling confident in preventing nocebo effects. The majority of respondents had not received formal training on placebo and nocebo effects, with most expressing an interest in further training in areas such as healthcare education, emphasizing communication skills to enhance placebo effects, and knowledge to recognize and reduce nocebo effects.

## Conclusions

There is a significant need for more comprehensive training on placebo and nocebo effects, particularly in early health professional education. These findings informed the development of educational resources and best practice recommendations developed as part of the outcomes from the PANACEA Consortium, improving the understanding and application of these effects among healthcare professionals across Europe.

## Introduction

Placebo effects represent positive changes in symptoms due to psycho-neuro-biological mechanisms activated by individual, psychological, and contextual factors whenever a patient engages with an active or non-active treatment, or enters a context of care [1]. Placebo effects concern the changes specifically attributable to placebo mechanisms. That is, the neurobiological and psychological mechanisms arising from patients experiencing symptom improvements purely from the expectation of care, the therapeutic environment, or interactions with healthcare providers. All these can significantly influence a patient's perception of health and recovery. Conversely, nocebo effects entail negative health changes driven by psychological and biological mechanisms, activated by individual and contextual factors [1–4].

Over the past two decades, research has advanced significantly in identifying the cognitive and emotional mechanisms underlying these effects [5,6]. Current evidence highlights how expectations (shaped by conditioning, observation, and verbal suggestion) and the psychosocial context of the treatment play a pivotal role in evoking placebo and nocebo effects [3,7–10].

Healthcare interactions, particularly the patient–provider relationship, can significantly influence health outcomes through the meanings and expectations conveyed during communication [11]. The clinician's role, behavior, and communication style can maintain, alter, or generate new expectations about treatment outcomes [12,13].

While the existing literature provides robust evidence supporting a significant influence of healthcare professional interactions on outcomes, it is important to acknowledge that some recent works, highlight ongoing debate about the precise role and impact of the healthcare provider in this process [14]. Understanding these dynamics and their influence on patient responses to treatment is essential for effectively enhancing positive and minimizing negative health outcomes.

Despite the advancements in the understanding of placebo and nocebo effects and their individual, clinical, psychological and contextual determinants, a range of attitudes among healthcare professionals persists [15,16]. Additionally, the lack of clarity in defining and applying the terms 'placebo' and 'placebo effects' raises important ethical considerations. The debate has shifted from questioning whether it is ethically acceptable to prescribe deceptive placebo treatment, to exploring how placebo interventions may be used without infringing on patient autonomy. By "placebo intervention," we generally refer to an approach that encompasses both tangible (such as pills or procedures) and contextual (such as the therapeutic environment and provider-patient interaction) factors that activate placebo effects, influencing patient responses through expectations and environmental cues. In this context, open-label placebos (i.e., honestly prescribed placebo interventions) have attracted growing interest, as several studies have shown either no superiority of deceptive placebos over open-label ones, or even non-inferiority. Nevertheless, deceptive placebo treatments remain common in clinical practice [17,18]. Healthcare professionals often navigate these ethical challenges while deliberately aiming to induce and manage placebo and nocebo effects based largely on personal experiences rather than sound scientific evidence [17,19].

This is particularly true concerning the extent to which tangible placebo interventions – whether pure (i.e., substances containing no active therapeutic ingredients, such as sugar pills, saline injections, or sham surgeries) or impure placebo (i.e., active treatments prescribed for their ability to activate placebo effects rather than for their direct pharmacological benefits)- can be administered without the patient's knowledge. In the U.S., physicians report frequently employing impure placebo interventions, such as antibiotics prescribed for viral infections, vitamin pills in the absence of a deficiency, or medications that are dosed too low to yield pharmacological effects [16,20,21]. Impure placebos are commonly used in the treatment of pain and functional disorders, and are often prescribed without explicitly informing the patient of their non-specific effects [22]. Although open-label placebos (OLP) have been proposed as a more ethical alternative that eliminates deception, debates continue regarding their efficacy and acceptability in clinical practice including the regard that OLP are perceived as disrespectful by some healthcare professionals [17,18,23,24].

Placebo interventions are common, but there is considerable variability in how they are defined, reported, and estimated [3,16]. This highlights the need to further investigate the motivations to use placebo interventions, particularly regarding non-specific therapies often employed to manage patient expectations and navigate clinical uncertainties, rather than solely to elicit placebo effects [20]. Reliance on anecdotal practice highlights the need for structured guidance and evidence-based strategies to harness these phenomena in clinical practice.

The PANACEA Consortium aims to advance understanding and clinical application of placebo and nocebo phenomena among European healthcare professionals. Although several surveys have focused on the motivations and attitudes towards placebo use among medical doctors and nurses [19], there is a notable lack of studies focused on the wider breadth of healthcare professionals. Moreover, surveys have primarily concentrated on placebo effects, with very few examining clinicians' understanding of nocebo effects or their experience with them. Additionally, to date, no survey has explored clinicians' interest in receiving training on these topics or their preferences regarding the format and content of such training.

This study builds on existing research by including a broader range of healthcare professionals, including medical professionals (e.g., family physicians, anesthesiologists, surgeons, neurologists) and nurses, as well as physiotherapists and psychologists. Moreover, this study extends prior research by not only considering placebo effects but also explicitly addressing nocebo effects, an area that has received comparatively little attention. Moreover, it goes beyond mapping knowledge and attitudes by also providing in-depth insights into healthcare professionals' preferences for training, thereby

offering concrete directions for the design and implementation of future educational programs. The findings will inform the development of educational resources to ensure the ethical and effective integration of placebo and nocebo effects into clinical practice.

## Materials and methods

### Study design and setting

This cross-sectional survey was conducted online using SurveyMonkey, a web-based survey platform, between December 2023 and January 2024. The survey is reported following the Checklist for Reporting Results of Internet E-Surveys (CHERRIES) [25]. The CHERRIES checklist is provided in S1 File. The survey was developed in accordance with COSMIN principles, which prioritize the use of validated instruments where available. We conducted a structured review of existing questionnaires on related topics [15,17,24–29], to identify relevant items and evaluate their alignment with our study objectives. Items judged to be relevant and psychometrically sound were incorporated, and additional close and open-ended questions were developed to address gaps not covered by existing tools. Prior to the launch, the survey was pre-tested with 45 healthcare professionals subscribed to the European Pain Federation EFIC mailing list to assess usability, technical functionality, acceptability, and refine question wording for clarity. Changes based on the feedback are included in S2 File.

### Ethics and consent

Ethical approval was obtained from the Bioethical Committee of the University of Bologna (protocol no. 0388068). The first page of the survey briefly described the study, its voluntary nature, and provided a link to the Participant Information Statement. Informed consent was obtained electronically via a checkbox before participants could proceed with the survey. All data were anonymized, and no personal identifiable information was collected.

### Participant inclusion criteria

Qualified healthcare professionals (self-reported) actively working in Europe from any healthcare discipline or specialty were included.

### Recruitment

Participants were recruited via email invitations sent through professional networks and posts on relevant social media groups, including the European Pain Federation (EFIC) online communication channels (webpage, quarterly newsletter, social media, LinkedIn), the Sant' Orsola Malpighi University Hospital of Bologna's dissemination channels, the mailing list of the College of Family Physicians in Poland, and the PANACEA Consortium's social media channels. As an incentive for participation, individuals who completed the survey in full could enter a raffle to win a one-year membership to the European Pain Federation EFIC Academy. Because recruitment was open through these channels, the number of individuals reached could not be determined, and therefore the formal response rate was not calculable.

### Survey administration

The survey was conducted in English and consisted of 46 questions across 10 pages/screens, featuring a mix of Likert scale, multiple-choice, and open-ended questions. The order of the questions was not randomized, and the topics were not counterbalanced, as the initial section aimed to gather participants' understanding of the placebo effects without explicitly using the term 'placebo'. This deliberate sequencing ensured that respondents provided unprompted knowledge and interpretations, allowing for a more accurate baseline assessment before introducing the term "placebo" [26]. The demographic data was collected in a dedicated section at the end of the survey, because this information was less central to the study objectives compared to the questions directly related to the research aims. This placement also aimed to

give priority to items closely aligned with the main subject of the study, increasing the likelihood that respondents would complete the most relevant questions. While participants were encouraged to complete all questions, this approach was intended to mitigate data loss for key variables if respondents chose to terminate the survey early. The survey was designed to take approximately 15 minutes to complete and was "open" (no log-in or password required). No formal control items (e.g., attention checks) were included in the survey to minimize respondent burden; instead, IP address marking was utilized to prevent multiple submissions, and addresses were not stored or utilized for any other purpose. No completeness checks were performed, and the respondents could not review or change their answers once submitted. Responses were not excluded based on the time taken to complete the survey and the full survey is available in S3 File.

## Survey outcomes

This study employed a mixed-methods approach, collecting both qualitative and quantitative data across five main sections.

The "Knowledge and Understanding" section aimed to gather insights into participants' definitions and perspectives on placebo and nocebo phenomena. This section included quantitative closed and open-ended questions to capture a comprehensive view of each respondent's ratings for *Perceived importance of factors influencing patient outcomes* (items 3–7); *Definitions of non-specific effects and placebo and nocebo phenomena* (items 8–11, 24); *Factors Contributing to Placebo and Nocebo Effects* (item 12); *Observation of Placebo and Nocebo Effects* (items 20–23, 26); and *Clinical Area Influenced by Placebo or Nocebo Effects* (items 16, 25).

The second section, "Applicability and Impact in Clinical Practice," included questions to quantify *Healthcare Providers' Use of Placebo Effects* (items 13–15, 17, 19) and *Confidence in Managing Placebo and Nocebo Effects* (items 27–30).

A separate "Ethical Acceptability" section addressed the ethical considerations surrounding the implementation of placebo in clinical practice using a series of closed-ended questions (items 18a-18f). The "Training and Educational Needs" section combined closed- and open-ended questions to quantify ongoing educational demands and gather qualitative feedback on personal development needs regarding placebo and nocebo (items 31–37).

Finally, the "Participant's Characteristics" section collected data on various demographic and professionals' attributes of the respondents such as country of residence, gender, age, profession, healthcare specialty, clinical setting, years of clinical experience, number of patients seen per week, and number of working hours per week (items 39–48).

**Data analysis.** Both complete and partial responses were included in the analyses. Responses were considered complete if all items of the survey were answered and partial if only a subset of items were completed. Partial responses were retained, as many participants completed substantial sections of the survey, and their data were considered valid for the items they answered. Data were analyzed using Stata, version 16.0 (StataCorp LLC). Descriptive statistics (means and standard deviations [SD], counts, and percentages) were used to summarize demographic and outcome data. A linear Pearson product-moment correlation analysis was conducted to examine the potential effect of participant fatigue on responses by analyzing the relationship between the order of item presentation and the number of responses. Logistic regression analysis was performed to determine whether responses to the first four questions predicted survey completion, with dropout versus continuing as the dichotomous outcome variable and demographic variables (sex, age, and profession) as independent variables. No adjustments (e.g., item weighting or propensity scores) were applied.

Free-text data were analyzed using Thematic Framework Analysis, whereby a five-step approach was utilized featuring (1) familiarization with the data, (2) development of a framework, (3) review and refinement of the framework, (4) coding of the data, and (5) charting of the data.

## Familiarization with the data

The data collected from the survey were exported to Microsoft Excel (including free-text responses). The first step included two of the authors (MOK and NS) reading and independently reviewing the free-text responses provided by participants. This step aimed to ensure a deep understanding and familiarity with the data and to identify any recurring

themes or patterns. NS, a musculoskeletal therapist (PhD, male), and MOK, a physiotherapist (PhD, female) both with experience in qualitative research with no previously established direct relationships with the respondents, conducted the qualitative component of this study.

### Developing a framework

Once familiar with the data, coding frameworks were independently developed by MOK and NS, and were structured around the key topics that emerged from the free-text responses of participants. The initial framework was created deductively (drawing from predefined concepts attributed to the questions) where concepts remained broad, serving as an initial guide for the development of additional structure.

### Review and refinement of the framework

The independently developed frameworks were then compared during a review to identify additional concepts. Any necessary modifications to the framework were discussed between MOK and NS to maintain maximum inclusivity. Conflicts were reassessed to ensure that the framework would comprehensively capture a variety of responses and insights. This step ensures reliability in the coding process by incorporating multiple independent perspectives of how the data should be interpreted and presented.

### Coding of the data

After the framework was finalized, MOK and NS proceeded to independently code all free-text responses according to predefined themes. This was done by systematically labeling the data with the corresponding themes from the framework. This enabled the data to be organized into manageable categories, making it easier to identify relationships and patterns between the themes. Codes which were of interest but did not correspond appropriately to a predetermined theme from the framework were coded as 'miscellaneous'. The themes were then compared, and conflicts were discussed and resolved by MOK and NS.

To strengthen the rigor of the process, MOK and NS work was subsequently re-evaluated independently and blinded by two additional authors, KM and AB.

### Charting the data

Following the coding process, the identified themes are presented narratively, noting the question and the themes that emerged from the responses to that question. Additional examples of supporting quotes for the qualitative survey responses are reported in S4 File.

## Results

### Respondents

A total of 807 individuals completed the consent form and began the survey, although the response rates for the individual questions varied considerably. There was a significant negative correlation between the order of the questions and the number of responses received ($r = -0.86$, $p < 0.001$), suggesting increased fatigue as the survey progressed. To ensure clarity and a comprehensive understanding of our findings, the results are organized and reported by each specific section of the survey. Readers are encouraged to consider the results in the context of each section's unique response dynamics, as this approach allows for a nuanced interpretation of the data, reflecting the varied engagement levels and insights provided by participants. The text or tables explicitly indicate the number of participants who responded to each survey section, providing clear insight into the response rates for each segment.

## Participant demographic data

212 participants provided demographic information at the conclusion of the survey. Among the 203 respondents who fully completed the survey, the average completion time was 30 minutes and seven seconds. The sample had a mean age of 44.8 years (SD = 14.4), with a balanced gender distribution: 104 females (49.06%), 105 males (49.53%), and three who preferred not to disclose their gender (1.41%). Nationality breakdown included the majority from Italy (117, 55.19%), followed by Poland (16, 7.55%), Sweden (12, 5.66%), the United Kingdom (11, 5.19%), Belgium (5, 2.36%), Spain (5, 2.36%), and Ireland (4, 1.89%), with 42 participants (19.81%) from other countries. Figs 1 and 2 provide additional details on participants' discipline, specialty, and clinical practice.

Within the self-reported 'Other specialty' category (n = 57, 17%), a wide range of medical specialties were provided. The most frequently reported were gynecology (n = 10) and emergency internal medicine (n = 7) and pain medicine (n = 5), followed by internal medicine (n = 5), clinical pharmacology and physiotherapy (n = 4). Emergency medicine and intensive care were each reported by three respondents. Specialties reported by two participants included anesthesiology and geriatrics. A wide variety of single-response other specialties were also recorded, such as dentistry, hematology, endocrinology, nephrology, otorhinolaryngology, midwifery, and palliative care. The low response rate for demographic data limited logistic regression analysis due to insufficient information for accurate predictions.

## Knowledge and understanding of placebo and nocebo effects

Respondents were asked to define factors that may influence treatment efficacy and outcomes through free-text responses to open-ended questions. In general, they described *non-specific effects* as outcomes not directly tied to a treatment's pharmacological action, but influenced by factors such as patient expectations and the patient-clinician relationship. These definitions were varied and seemed to be influenced by discipline-specific perspectives. Psychologists

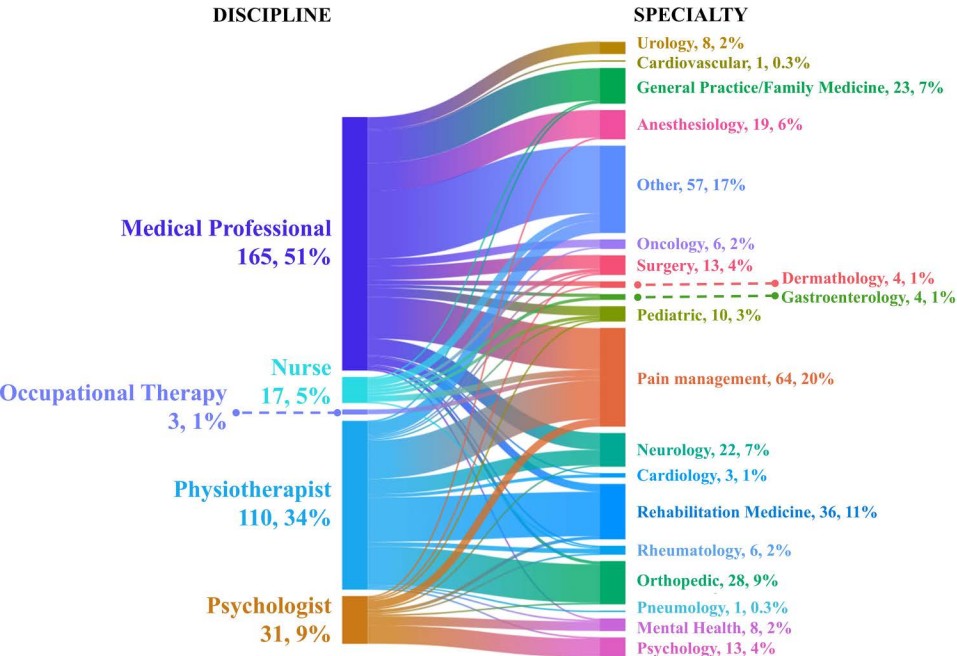

**Fig 1. Participant characteristics by discipline and specialty.** *Note.* The "Other" category in Discipline is not displayed in the figure (11 answers, 5.19%). Participants could select more than one response; therefore, the percentages reported reflect the frequency of responses, and the total may exceed 100%. A total of 212 (26.3%) participants answered this question.

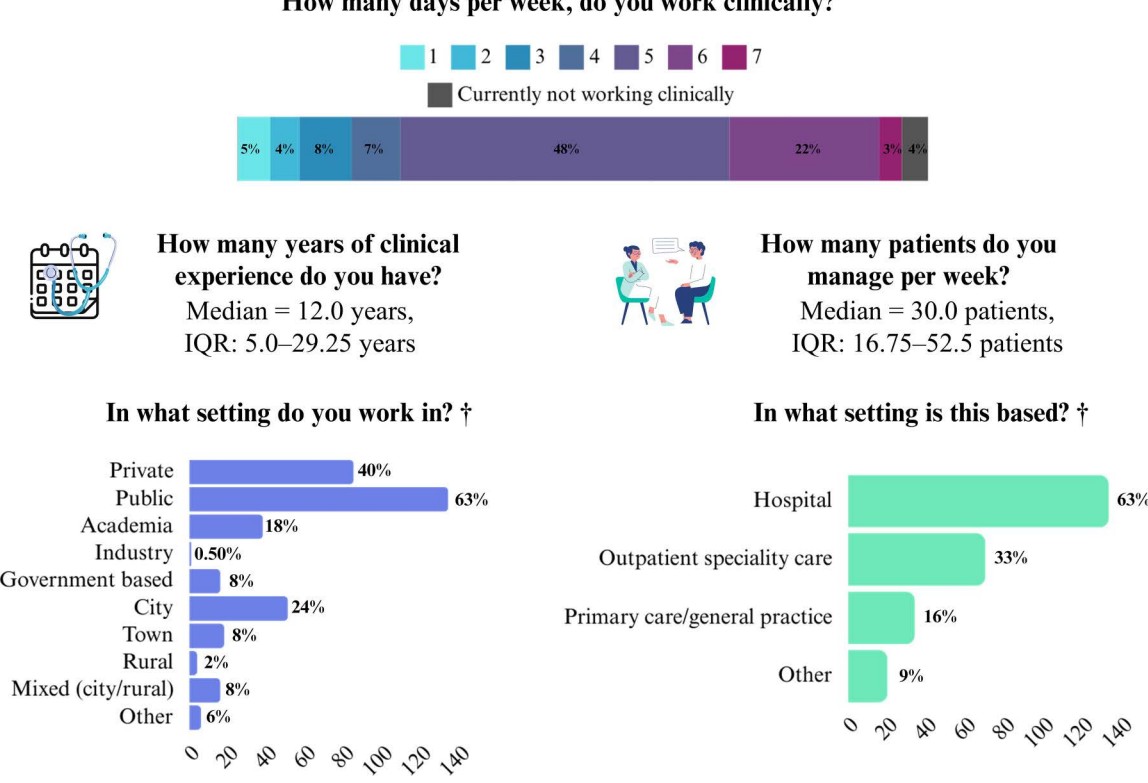

**Fig 2. Participants' information about their clinical practice.** *Note.* †Participants could select more than one response; therefore, the percentages reported reflect the frequency of responses, and the total may exceed 100%. A total of 212 (26.3%) participants answered this question.

emphasized that contextual elements such as empathy and communication are key, noting, "Good therapeutic alliance; patient trusts therapist and feels they are interested; positive expectations for both therapist and patient" [*Female, age 70, Psychology*]. Medical professionals highlighted trust in treatment or clinician authority, viewing non-specific effects as outcomes influenced by beliefs and unintended consequences. Physiotherapists focused on contextual and environmental factors, such as "The relaxation, empathy, and security that the subject perceives" [*Male, age 50, Physiotherapy*], however nurses adopted a practical lens, describing them as "an outcome or change in a patient… not directly due to the treatment" [*Female, age 45, Nursing,*]

*Placebo effects* were viewed as improvements driven by patient beliefs independent of the treatment's active properties. Across disciplines, responses highlighted the role of expectations, trust, and therapeutic context in shaping these effects. Psychologists emphasized the power of belief and positive expectations, stating, "Positive outcomes attributed to being a part of/being provided an intervention, regardless of what actually is provided" [*Female, age 30, Psychology*]. Medical professionals focused on patient confidence in treatment and the clinician, noting, "Placebo effects mean beneficial outcome in patient's treatment. It is also doctor and patient's confidence" [*Female, age 35, Medical professional*]. Nurses linked placebo effects to imagined benefits, describing them as "the effect that a patient imagines they are getting from a particular treatment" [*Female, age 45, Nursing*]. Occupational therapists and physiotherapists highlighted communication and empowerment, explaining how beliefs in treatment foster positive outcomes.

*Placebo treatments* were seen as leveraging trust and clinician interaction to enhance outcomes, although ethical concerns emerged regarding their use. Medical professionals expressed conflict, "To be honest, for me it's a lie to the

patient… Otherwise, it's a lie" [*Male, age 32, Medical Professional*], reflecting discomfort with deception. Psychologists and physiotherapists emphasized fostering trust and meeting patient needs, describing placebo treatments as "non-evidence-based techniques deliberately used to induce the placebo effect" [*Male, age 40, Physiotherapy*]. Nurses often took a pragmatic view, such as using "a treatment with no active component" [*Female, age 45, Nursing*], thus framing placebo treatments as ways to reassure patients.

Conversely, *nocebo effects* were linked to negative expectations and poor communication. Medical professionals highlighted the role of communication failures, with one noting, "Nocebo effects reveal usually doctor-patient's bad communication" [*Female, age 35, Medical Professional*]. Psychologists emphasized the impact of negative expectations, noting that "adverse expectations might contribute to experiencing negative symptoms" [*Female, age 40, Psychology*]. Participants across all disciplines pointed to the influence of negative language and a lack of empathy, emphasizing the need for strategies to prevent these responses. One advised that, "[...] to minimize the negative effects, [one must take] care of the attitudes that drive to nocebo effects" [*Rather not say, age 61, Medical Professional*].

Challenges related to nocebo effects were also underscored, where negative expectations were seen as exacerbating symptoms despite the harmless nature of the treatments. Respondents identified triggers such as poor communication, negative framing, and external influences such as misinformation from online sources. From their experiences, participants described cases of heightened side effects due to patient anxiety and increased pain associated with negative prior experiences.

Survey data revealed that 47.7% ($n = 113$) of respondents felt somewhat confident in recognizing symptom improvement due to placebo effects, with only 0.8% ($n = 2$) feeling extremely confident. In contrast, 55.8% ($n = 125$) had observed nocebo effects in their clinics, and 54.9% ($n = 123$) believed that this occasionally influenced patients' health. Confidence in recognizing nocebo effects was low, with only 8.0% ($n = 18$) feeling very confident and 0.9% ($n = 2$) feeling extremely confident. Further details regarding knowledge and understanding of placebo and nocebo effects are provided in Table 1.

There was significant variation in perceptions of which clinical disciplines may benefit from placebo effects or for which clinical conditions nocebo effects are prone. Placebo was believed to benefit clinical disciplines like alternative medicines ($n = 143$, 60.3%), psychology ($n = 143$, 60.3%), and general medicine ($n = 129$, 54.4%). Psychiatry ($n = 125$, 55.8%), psychology ($n = 113$, 50.5%), and general medicine ($n = 113$, 50.5%) were believed to be most prone to nocebo effects. Further details are provided in Fig 3.

## Applicability and impact in clinical practice

The results for questions regarding the perception of contribution of placebo and nocebo effects in clinical practice, alongside reported confidence in managing these phenomena, are summarized in Tables 2 and 3, which also outline the frequency of treatments believed to activate placebo effects or enhance their impact through routine clinical interactions.

Quantitative data indicated that 71.7% ($n = 170$) of respondents reported taking advantage of placebo effects in their clinical practice. Participants noted a significant positive impact of placebo effects on patient outcomes "It helps to achieve best possible outcomes" [*Female, age 50, Gynecologic Oncology*], and emphasized the influence of trust "A easy way to take the trust/confidence of the patient" [*Male, age 26, Physiotherapy*], positive framing "Yes, by trying to create positive expectation, to provide reassurance that the treatment will help, "positive framing" [*Female, age 58, Medical professional*], and therapeutic relationships "they [*placebo*] are part of engaging the patient, building hope and positive expectation, developing a trusting relationship" [*Female, age 70, Psychology*]. Many highlighted the role of placebo effects in enhancing compliance "Because patients who have a good relationship with the surgeon have a better compliance" [*Male, age 61, Medical professional*], and fostering hope "to build a good relation and instill hope" [*Female, age 47, Psychology*], with aims of improving symptoms and well-being. Others viewed it as a valuable complement to active treatments, particularly in conditions influenced by psychological factors. Examples include the use of saline solutions to effectively manage pain "After each [*saline*] infusion, the pain intensity is assessed, that way we can identify if patients with potential strong

**Table 1. Knowledge and understanding of placebo and nocebo effects.**

| *Perceived importance of factors influencing patient outcomes* 569/807 (70.5%) | **Extremely important** (*n*, %) | **Very important** (*n*, %) | **Somewhat important** (*n*, %) | **Not so important** (*n*, %) | **Not at all important** (*n*, %) |
|---|---|---|---|---|---|
| Healthcare professional conveying positive expectations in patients (Item 3) | 218 (38.3%) | 298 (52.4%) | 50 (8.8%) | 2 (0.3%) | 1 (0.2%) |
| Patient believing in the treatment (Item 4) | 278 (48.9%) | 248 (43.6%) | 40 (7.0%) | 1 (0.2%) | 2 (0.4%) |
| Patient having positive past experiences with the treatment (Item 5) | 182 (32.0%) | 264 (46.4%) | 102 (17.9%) | 19 (3.3%) | 2 (0.4%) |
| Patient trusting in the healthcare professional (Item 6) | 294 (51.7%) | 248 (43.6%) | 27 (4.8%) | 0 | 0 |
| Healthcare professional informing clearly and checking the patient's comprehension about the treatment (Item 7) | 337 (59.2%) | 202 (35.5%) | 27 (4.8%) | 3 (0.5%) | 0 |
| *Perceived contributors to the placebo effect* (Item 12) 237/807 (29.4%) | **Strongly Disagree** (*n*, %) | **Somewhat Disagree** (*n*, %) | **Neutral** (*n*, %) | **Somewhat Agree** (*n*, %) | **Strongly Agree** (*n*, %) |
| A treatment with no active ingredient | 25 (10.6%) | 19 (8.0%) | 42 (17.7%) | 80 (33.8%) | 71 (30.0%) |
| A treatment with an active ingredient but not specific for the clinical condition | 31 (13.1%) | 33 (13.9%) | 62 (26.2%) | 88 (37.1%) | 23 (9.7%) |
| An aspect related to the clinical context | 6 (2.5%) | 12 (5.1%) | 37 (15.6%) | 111 (46.8%) | 71 (30.0%) |
| An aspect related to patient-clinician relationship | 10 (4.2%) | 12 (5.1%) | 23 (9.7%) | 83 (35.0%) | 109 (46.0%) |
| Patient expectations | 12 (5.1%) | 7 (3.0%) | 20 (8.4%) | 68 (28.7%) | 130 (54.9%) |
| Patient past experience | 10 (4.2%) | 8 (3.4%) | 27 (11.4%) | 106 (44.7%) | 86 (36.3%) |
| A non-specific effect | 11 (4.6%) | 20 (8.4%) | 87 (36.7%) | 87 (36.7%) | 32 (13.5%) |
| Regression to the mean | 15 (6.3%) | 27 (11.4%) | 101 (42.6%) | 71 (30.0%) | 23 (9.7%) |
| Natural disease progression | 24 (10.1%) | 35 (14.8%) | 72 (30.4%) | 77 (32.5%) | 29 (12.2%) |
| | **Not at all confident** (*n*, %) | **Not so confident** (*n*, %) | **Somewhat confident** (*n*, %) | **Very confident** (*n*, %) | **Extremely confident** (*n*, %) |
| *How confident do you feel in recognizing when symptom improvement is due to placebo effects?* (Item 20) 237/807 (29.4%) | 24 (10.1%) | 74 (31.2%) | 113 (47.7%) | 24 (10.1%) | 2 (0.8%) |
| *How confident do you feel in recognizing when symptom worsening/deterioration is due to nocebo effects?* (Item 26) 224/807 (27.8%) | 34 (15.2%) | 69 (30.8%) | 101 (45.1%) | 18 (8.0%) | 2 (0.9%) |

*Note.* For each main item (bold and italic), the number of respondents is reported alongside the total number of survey participants. For each answer option, the number of respondents is reported, along with the percentage relative to the total number of answer options.

placebo response" [*Female, age 55, Medical professional*], driven by positive patient expectations. Ethical considerations were occasionally discussed, with respondents sometimes prioritizing patient transparency and well-being "to explore psychological impacts, always with transparency and the patient's well-being in mind." [*Female, age 50, Gynecologic Oncology*].

## Ethical acceptability

Table 4 provides insights into respondents' perspectives on the use of treatments, with and without patient consent. Generally, placebo use with consent was considered ethical; however, placebo use without consent was largely deemed unethical, with nearly 50% (*n* = 110) of respondents indicating 'never' or 'rarely' ethical. A similar pattern is observed for substances with no direct therapeutic effects, where patient consent plays a significant role in the perceived ethical acceptability.

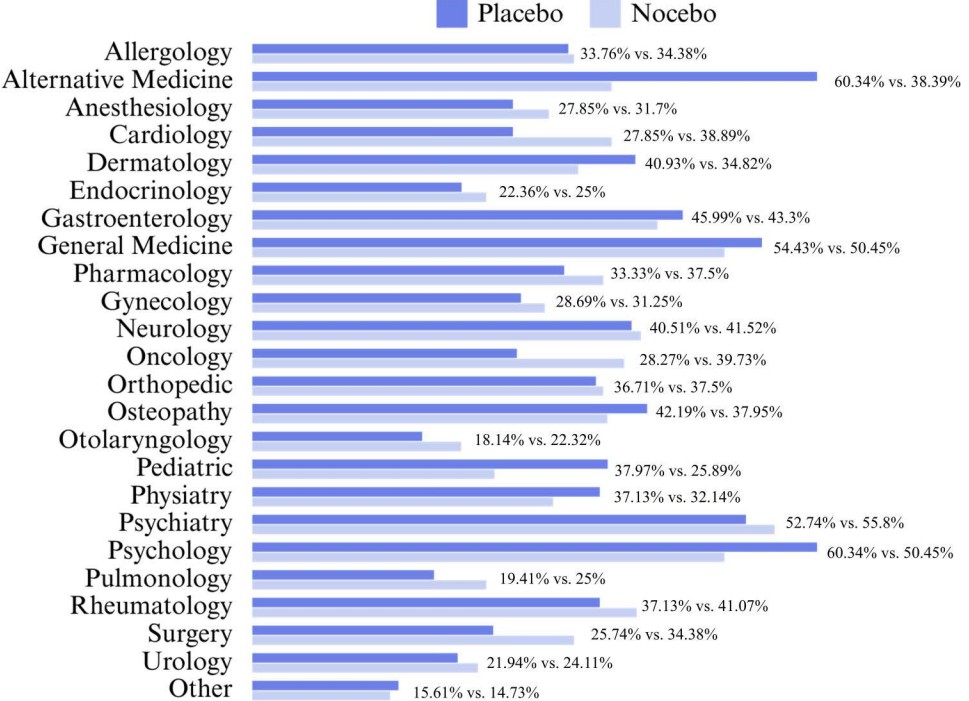

**Fig 3. Specialties in which clinical conditions may benefit from placebo effects or be prone to nocebo effects: respondents' experiences.**
*Note.* A total of 237 respondents answered the question about placebo effects (item 16: "Please select all the specialties in which clinical conditions, in your experience, may benefit from placebo effects.", while 224 respondents answered the question about nocebo effects (item 25: "Please select all the specialties in which clinical conditions, in your experience, may be prone to nocebo effects.").

## Training and educational needs

Results showed that 63.1% (*n* = 137) of the respondents had not received formal training on placebo effects, 67.7% (*n* = 147) indicated no formal training on nocebo effects, and 9.7% (*n* = 21) were unsure. Interest in further training was notable, with 83.9% (*n* = 182) expressing being at least 'somewhat interested' in learning about placebo effects and 86.2% (*n* = 187) regarding nocebo effects. The training methods perceived as most helpful in further placebo and nocebo training are detailed in Table 5, which suggests self-study (57.1%, *n* = 88), skills workshops (56.7%, *n* = 84), and clinical placements (50.4%, *n* = 73) as the methods with the most ratings for the answer options 'somewhat' or 'extremely' helpful.

Qualitative data, summarized in the next section, highlighted the need for comprehensive education on placebo and nocebo mechanisms, emphasizing their significance in clinical practice. There was a need for training across all disciplines that extended beyond theoretical knowledge to include practical applications, such as strategies to ethically harness placebo effects through communication.

A physiotherapist expressed a desire to "learn more about the placebo mechanism and influence on patient outcome" [*Male, age 38, Physiotherapy*], emphasizing the need to understand its psychological and neurobiological underpinnings. Ethical considerations emerged as a theme across disciplines, with a nurse highlighting the need to address "ethical implications, legal implications, and when and how we are allowed to use [placebo and nocebo effects]" [*Male, age 25, Nursing*]. Psychologists similarly stressed the importance of navigating ethical challenges in placebo use, particularly in psychotherapy settings where trust and communication are critical. Medical professionals echoed these concerns, emphasizing that ethical considerations must accompany practical applications in areas like oncology, surgery, and pain management. Respondents across all fields emphasized the need for ethical training with clear guidelines to maintain patient

**Table 2. Healthcare providers' use of placebo effects.**

| *To what extent does the following treatment contribute to activating the placebo effect?* (Item 17) 237/807 (29.4%) | **Never** (*n*, %) | **Rarely** (*n*, %) | **Neutral** (*n*, %) | **Sometimes** (*n*, %) | **Always** (*n*, %) |
|---|---|---|---|---|---|
| Vitamins | 9 (3.8%) | 13 (5.5%) | 45 (19.0%) | 145 (61.2%) | 25 (10.6%) |
| Supplements | 8 (3.4%) | 13 (5.5%) | 41 (17.3%) | 152 (64.1%) | 23 (9.7%) |
| Antidepressants | 17 (7.2%) | 15 (6.3%) | 56 (23.6%) | 121 (51.1%) | 28 (11.9%) |
| Anxiolytics | 16 (6.8%) | 16 (6.8%) | 45 (19.0%) | 129 (54.4%) | 31 (13.1%) |
| Antibiotics | 51 (21.5%) | 52 (21.9%) | 67 (28.3%) | 59 (24.9%) | 8 (3.4%) |
| Painkillers | 16 (6.8%) | 10 (4.2%) | 42 (17.7%) | 131 (55.3%) | 38 (16.0%) |
| Physiotherapeutic treatments | 10 (4.2%) | 11 (4.6%) | 33 (13.9%) | 143 (60.3%) | 40 (16.9%) |
| Instrumental treatments | 24 (10.1%) | 26 (11.0%) | 49 (20.7%) | 118 (49.8%) | 20 (8.4%) |
| Surgery | 45 (19.0%) | 45 (19.0%) | 43 (18.1%) | 89 (37.6%) | 15 (6.3%) |
| Probiotics | 13 (5.5%) | 13 (5.5%) | 70 (29.5%) | 121 (51.1%) | 20 (8.4%) |
| *To what extent do you agree with the following reasons for providing placebo treatments?* (Item 19) 237/807 (29.4%) | **Strongly Disagree** (*n*, %) | **Somewhat Disagree** (*n*, %) | **Neutral** (*n*, %) | **Somewhat Agree** (*n*, %) | **Strongly Agree** (*n*, %) |
| Genuine benefits | 13 (5.5%) | 9 (3.8%) | 37 (15.6%) | 110 (46.4%) | 68 (28.7%) |
| Patient wants or expects a treatment | 32 (13.5%) | 26 (11.0%) | 47 (19.8%) | 104 (43.9%) | 28 (11.8%) |
| No treatment exists | 17 (7.2%) | 21 (8.9%) | 42 (17.7%) | 116 (49.0%) | 41 (17.3%) |
| Adjunct to active treatment | 16 (6.8%) | 20 (8.4%) | 28 (11.8%) | 120 (50.6%) | 53 (22.4%) |
| To diagnose psychosomatic symptoms | 37 (15.6%) | 28 (11.8%) | 44 (18.6%) | 82 (34.6%) | 46 (19.4%) |
| To treat medically unexplained symptoms | 37 (15.6%) | 43 (18.1%) | 52 (21.9%) | 87 (36.7%) | 18 (7.6%) |
| To diagnose patient malingering | 53 (22.4%) | 37 (15.6%) | 72 (30.4%) | 52 (21.9%) | 23 (9.7%) |
| For palliative therapies | 33 (13.9%) | 19 (8.02%) | 59 (24.9%) | 81 (34.2%) | 45 (18.9%) |

*Note.* For each main item (bold and italic), the number of respondents is reported alongside the total number of survey participants. For each answer option, the number of respondents is reported, along with the percentage relative to the total number of answer options.

**Table 3. Confidence in managing placebo and nocebo effects.**

| | **Not at all confident** (*n*, %) | **Not so confident** (*n*, %) | **Somewhat confident** (*n*, %) | **Very confident** (*n*, %) | **Extremely confident** (*n*, %) |
|---|---|---|---|---|---|
| *To what extent are you confident in... harnessing placebo effects in your clinical practice?* (Item 27) 217/807 (26.9%) | 18 (8.3%) | 57 (26.3%) | 116 (53.5%) | 23 (10.6%) | 3 (1.4%) |
| *preventing nocebo effects in your clinical practice?* (Item 28) 217/807 (26.9%) | 20 (9.2%) | 65 (30.0%) | 103 (47.5%) | 28 (12.9%) | 1 (0.5%) |
| *communicating about placebo effects in your clinical practice?* (Item 29) 217/807 (26.9%) | 20 (9.2%) | 49 (22.6%) | 97 (44.7%) | 47 (21.7%) | 4 (1.8%) |
| *communicating about nocebo effects in your clinical practice?* (Item 30) 217/807 (26.9%) | 26 (12.0%) | 59 (27.2%) | 79 (36.4%) | 46 (21.2%) | 7 (3.2%) |

*Note.* For each main item (bold and italic), the number of respondents is reported alongside the total number of survey participants. For each answer option, the number of respondents is reported, along with the percentage relative to the total number of answer options.

**Table 4. Ethical acceptability.**

| *How ethical are the following?*<br>237/807 (29.4%) | Never<br>(*n*, %) | Rarely<br>(*n*, %) | Neither<br>(*n*, %) | Sometimes<br>(*n*, %) | Always<br>(*n*, %) |
|---|---|---|---|---|---|
| Using placebo effects with consent (Item 18a) | 11 (4.6%) | 6 (2.5%) | 32 (13.5%) | 64 (27.0%) | 124 (53.3%) |
| Using placebo effects without consent (Item 18b) | 55 (23.2%) | 50 (21.1%) | 52 (21.9%) | 70 (29.5%) | 10 (4.2%) |
| Using inert substances with consent (Item 18c) | 11 (4.6%) | 15 (6.3%) | 51 (21.5%) | 75 (31.7%) | 85 (35.9%) |
| Using inert substances without consent (Item 18d) | 86 (36.3%) | 64 (27.0%) | 44 (18.6%) | 41 (17.3%) | 2 (0.8%) |
| Using substances or methods which have a known pharmacological or physical activity but which cannot be expected to have any direct therapeutic effects for the respective disease and in the chosen dosage with consent (Item 18e) | 20 (8.4%) | 25 (10.6%) | 53 (22.4%) | 75 (31.7%) | 64 (27.0%) |
| Using substances or methods which have a known pharmacological or physical activity but which cannot be expected to have any direct therapeutic effects for the respective disease and in the chosen dosage without consent (Item 18f) | 98 (41.4%) | 60 (25.3%) | 42 (17.7%) | 35 (14.8%) | 2 (0.8%) |

Note. For each main item (bold and italic), the number of respondents is reported alongside the total number of survey participants. For each answer option, the number of respondents is reported, along with the percentage relative to the total number of answer options.

trust, particularly in fields such as pain management, psychiatry, and neurology, where psychological factors significantly influence treatment outcomes.

The gap in training on recognizing and managing nocebo effects was emphasized across disciplines, highlighting a range of training needs for improving understanding, practical strategies, and communication techniques to mitigate the impact of nocebo effects. Respondents pointed out the importance of understanding triggers, managing patient expectations, and employing effective communication. One respondent highlighted the need for training on "how to minimize the negative effects, taking care of the attitudes that drive to nocebo effects." [*Rather not say, age 61, Medical Professional*]. The proposed solutions included clinical scenarios, communication workshops, and practical examples (see Table 5).

## Discussion

This study examined a European multi-country convenience sample of healthcare professionals' knowledge, practices, and training needs regarding placebo and nocebo effects. Although our findings reveal healthcare

**Table 5. Perceived helpfulness of previous training methods.**

| *Helpfulness of previous training methods* (Item 33)<br>160/807 (19.8%) | Extremely unhelpful<br>(*n*, %) | Somewhat unhelpful<br>(*n*, %) | Neutral<br>(*n*, %) | Somewhat helpful<br>(*n*, %) | Extremely helpful<br>(*n*, %) |
|---|---|---|---|---|---|
| Clinical placement | 10 (6.9%) | 12 (8.3%) | 50 (34.5%) | 51 (35.2%) | 22 (15.2%) |
| Self-study | 18 (11.7%) | 6 (3.9%) | 42 (27.3%) | 73 (47.4%) | 15 (9.7%) |
| Skills workshops | 14 (9.5%) | 7 (4.7%) | 43 (29.1%) | 60 (40.5%) | 24 (16.2%) |
| Apps | 18 (3.1%) | 17 (12.4%) | 72 (52.6%) | 25 (18.3%) | 5 (3.7%) |
| Self-study workbooks | 14 (9.8%) | 19 (13.2%) | 51 (35.7%) | 54 (37.8%) | 5 (3.5%) |
| College training courses | 16 (11.4%) | 8 (5.7%) | 49 (34.8%) | 53 (37.6%) | 15 (10.6%) |
| University modules | 16 (11.2%) | 9 (6.3%) | 48 (33.6%) | 50 (35.0%) | 20 (14.0%) |

Note. For each main item (bold and italic), the number of respondents is reported alongside the total number of survey participants. For each answer option, the number of respondents is reported, along with the percentage relative to the total number of answer options.

professionals have some understanding of placebo and nocebo effects and their potential impact in clinical practice, there remain significant gaps in knowledge and training regarding placebo and nocebo effects. In consideration of the role of placebo and nocebo mechanisms on clinical outcomes [1], these results highlight the urgent need to incorporate comprehensive, evidence-based education on placebo and nocebo effects into healthcare curricula.

In line with previous studies [16,19,27,28], the high prevalence of placebo interventions in clinical practice (71.7% of respondents) highlights the implicit recognition of its potential to enhance treatment outcomes. The clinical domains most frequently reported that may benefit from placebo effects were general medicine, neurology, psychiatry, gynecology, cardiology, gastroenterology, and dermatology. This aligns with the wide range of conditions under which placebo effects have been documented [1,29].

Nevertheless, notable variations in the understanding and application of these effects were observed across different disciplines and clinical areas. Respondents generally attribute placebo effects to patient beliefs independent of specific treatment properties, emphasizing their perceived utility, especially in conditions lacking specific treatments. However, this perspective may reflect an underestimation of the healthcare professional's own role in influencing placebo effects through communication and interaction, as similarly observed by Druart et al (2023) [16], highlighting the need for greater awareness of how their behavior and communication strategies can shape patient outcomes.

While many reported taking advantage of placebo effects, the understanding of the underlying mechanisms varied significantly across disciplines. Psychologists highlighted the roles of beliefs and positive expectations, while medical professionals emphasized patient trust in their healthcare provider. Nurses associated placebo effects with imagined benefits, while occupational therapists and physiotherapists emphasized communication and empowerment as factors fostering positive outcomes. Conversely, nocebo effects were frequently linked to poor clinician-patient communication or misinformation from unreliable online sources. This variability aligns with previous research on the diverse understanding and application of placebo effects across professions [16,19,27,28].

Variability in the understanding of placebo effects extends to their use in practice. While many healthcare professionals frequently reported attempting to harness placebo effects through treatment framing, treatment adjustments, drug dosage modifications or a range of impure placebos (most commonly vitamins, supplements, anxiolytics, and physiotherapy), confidence in leveraging these effects was variable. Respondents reported generally low confidence in harnessing placebo effects and preventing nocebo effects, highlighting a clear gap between awareness of these phenomena and the ability to apply them effectively in practice. This gap underscores the importance of targeted education focusing on practical strategies, ethical considerations, and communication skills.

Our findings also highlight the complex ethical considerations surrounding placebo use, a theme consistently addressed in the literature [8,30,31]. While many clinicians consider placebo treatments ethically permissible, concerns are reflected in these results regarding deception of patients, particularly in the use of impure placebos. This practice was also noted in a study on Swiss primary care providers [32]. In our survey, the majority (79.3%) found placebo use to be acceptable with patient consent; however, this acceptance dropped to 33.7% without consent. This difference underscores the importance of open communication in managing patient expectations and mitigating the risk of nocebo effects [33–36]. The sensitive ethical dimensions involved demand careful consideration and clear guidelines for clinical practice. In addition, placebo and nocebo responses may be triggered unintentionally through routine communication and care, even without provider awareness. This underscores the importance of ensuring all healthcare professionals receive foundational training to recognize and manage these effects.

Lastly, our findings reveal substantial gaps in current placebo and nocebo effects training. A significant proportion of respondents (63.1% for placebo, 67.7% for nocebo) reported receiving no formal training, a deficiency compounded by low confidence levels in managing these effects (only 12% and 13.4% were either very or extremely confident in

harnessing placebo and nocebo effects, respectively). These findings align with previous research, highlighting the lack of structured education on placebo and nocebo mechanisms in medical curricula, despite growing recognition of their clinical relevance [15]. Literature has emphasized the need for improved training and provided clinical recommendations to bridge this gap, including competency-based learning approaches to enhance communication skills and patient outcomes [37]. Respondents' feedback highlighted the critical need for comprehensive training in three key areas: (1) effective communication strategies for managing both placebo and nocebo effects, (2) ethical considerations in the application of these effects, and (3) a deeper understanding of the underlying mechanisms. Future research should examine whether training is introduced during initial qualifications or later in professional development, to inform how best to embed these topics across career stages. Ideally, such training would begin early in professional education and include practical skills training using methods such as clinical scenarios, simulations, and case studies.

### Limitations and strengths

This study offers valuable insights into healthcare professionals' understanding and application of placebo and nocebo effects; however, its limitations should be acknowledged. The response rate varied significantly across survey sections, potentially introducing bias and limiting the generalizability of some findings. Considering the overrepresentation of Italian participants (55.19%), and pain professionals (31.6%) and such a broad potential population base, overall participation was low, and several large countries, with active placebo researchers and large populations, were sparsely represented. Notably, out of 807 participants who initiated the survey, we have complete demographic data (last session) for only 212 respondents who fully completed the survey, limiting our ability to ascertain the precise representation of European countries within the sample. This underrepresentation limits the extrapolation of findings to a wider European context and reflects the tendency for main participating countries to align with the affiliations of study authors. Additionally, potential threats to internal validity warrant consideration. Because the sequence of questions was not randomized, we cannot exclude the possibility of order or priming effects that may have influenced participants' responses. This design choice reflected a pragmatic decision to prioritize items most closely aligned with the main aims of the study, with the intention of maximizing completion of the most relevant questions. Furthermore, the cross-sectional design of the current study prevents the establishment of causal relationships between knowledge, attitudes, and practices. Alongside an absence of formal attention checks and a reliance on self-reported data introducing a potential response bias, the survey was provided in English language only limiting responses from non-English speakers. The high proportion of partial completions also represents a limitation. Nevertheless, these responses were included in analyses because many participants completed substantial sections of the survey, and their data were considered valid for the items answered. In addition, while the findings captured genuine insights within our sample, their transferability across the broader European healthcare workforce should be considered appropriately, as other factors (e.g., age, clinical setting) may also play a role. Finally, it should be acknowledged that the type of treatment delivered by different healthcare professionals may shape how placebo and nocebo effects are perceived and manifested. For example, side effects ascribed to a pharmacological treatment may be experienced differently than those emerging in psychological interventions. This disciplinary variability highlights the complexity of interpreting our findings across professional groups.

Despite these limitations, the study possesses several considerable strengths. Although the participants were not evenly distributed across countries, the inclusion of an international sample enhanced the generalizability of the quantitative findings and broadened the scope beyond studies confined to single nations. The mixed-methods approach, integrating quantitative and qualitative data, provided a rich and nuanced understanding of healthcare professionals' perspectives. A detailed exploration of healthcare professionals' ethical considerations surrounding placebo and nocebo effects adds significant value to existing literature. Finally, the rigorous methodology, adhering to the CHERRIES checklist, ensured the reliability and transparency of our findings.

## Conclusion

This study mapped the current knowledge and identified significant gaps in healthcare professionals' understanding and management of placebo and nocebo effects, emphasizing the critical need for improved training and education. The observed variations in knowledge and application across disciplines and low reported confidence levels highlight the need for targeted interventions. Future research should focus on developing and evaluating evidence-based training programs that address these gaps and promote the ethical and effective use of placebo and nocebo effects to optimize patient outcomes. This research should utilize the feedback provided by healthcare professionals to inform the design of these interventions.

## Supporting information

**S1 File. Supplement PO.**
(DOCX)

**S2 File. Raw Data.**
(XLSX)

**S3 File. Inclusivity in global research questionnaire.**
(DOCX)

## Acknowledgments

The authors thank Melinda Borzsak-Schramm and Sam Kynman (EFIC's Executive Office) for their organizational and recruitment support.

This manuscript was prepared on behalf of the *PANACEA Consortium*. The members of the PANACEA Consortium are: Katia Mattarozzi (lead author), Arianna Bagnis, Barbara Vetturini from the University of Bologna; Mary O'Keeffe, Nathan Skidmore, Sam Kynman, and Melinda Borzsak-Schramm from the European Pain Federation EFIC; Stefanie Meeuwis and Andrea Evers from Leiden University; Julia Haas, Eveliina Glogan, Johan Vlaeyen, Lidia Guadagnoli, and Tom Beckers from Katholieke Universiteit Leuven; Elżbieta Bajcar, Przemysław Babel, Marek Oleszczyk, and Agata Stalmach Przygoda from Jagiellonian University; Antonio Portoles and Ana Belen Rivas from Universidad Complutense Madrid.

## Author contributions

**Conceptualization:** Mary O'Keeffe, Arianna Bagnis, Katia Mattarozzi.

**Data curation:** Mary O'Keeffe, Nathan Skidmore.

**Formal analysis:** Mary O'Keeffe, Nathan Skidmore, Arianna Bagnis, Katia Mattarozzi.

**Funding acquisition:** Arianna Bagnis, Katia Mattarozzi.

**Investigation:** Mary O'Keeffe, Nathan Skidmore.

**Methodology:** Mary O'Keeffe, Nathan Skidmore, Arianna Bagnis, Przemysław Bąbel, Elżbieta A. Bajcar, Alessandra De Palma, Andrea W.M. Evers, Eveliina Glogan, Julia W. Haas, Stefanie H. Meeuwis, Marek Oleszczyk, Antonio Portolés, Johan W.S. Vlaeyen, Katia Mattarozzi.

**Project administration:** Mary O'Keeffe, Arianna Bagnis, Katia Mattarozzi.

**Supervision:** Mary O'Keeffe, Arianna Bagnis, Katia Mattarozzi.

**Validation:** Mary O'Keeffe, Nathan Skidmore, Arianna Bagnis, Katia Mattarozzi.

**Visualization:** Nathan Skidmore, Arianna Bagnis.

**Writing – original draft:** Mary O'Keeffe, Nathan Skidmore.

**Writing – review & editing:** Mary O'Keeffe, Nathan Skidmore, Arianna Bagnis, Przemysław Bąbel, Elżbieta A. Bajcar, Alessandra De Palma, Andrea W.M. Evers, Eveliina Glogan, Julia W. Haas, Stefanie H. Meeuwis, Marek Oleszczyk, Antonio Portolés, Johan W.S. Vlaeyen, Katia Mattarozzi.

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
