## [Decision Letter · Decision Letter 0]

27 Aug 2025

Dear Dr. Bagnis,

Thank you for submitting your manuscript to PLOS ONE. After careful consideration, we feel that it has merit but does not fully meet PLOS ONE’s publication criteria as it currently stands. Therefore, we invite you to submit a revised version of the manuscript that addresses the points raised during the review process.

However, Reviewers raised comments that need to be addressed. So, I like to invite the authors to revise their manuscript and resubmit it for further evaluation properly.

We look forward to receiving your revised manuscript.

Kind regards,

Vilfredo De Pascalis

Academic Editor

PLOS ONE

Journal Requirements:

The study is funded by a grant from the Erasmus+ Program of the European Union (grant number: IT02-KA220-HED- 000088065)

M. O’Keeffe is supported by an Irish fellowship (UCD Ad Astra Fellowship).

6. Please amend the manuscript submission data (via Edit Submission) to include author Nathan Skidmore and Antonio Portolés

7. Please amend your authorship list in your manuscript file to include author Nathaniel Skidmore

8. One of the noted authors is a group or consortium PANACEA Consortium. In addition to naming the author group, please list the individual authors and affiliations within this group in the acknowledgments section of your manuscript. Please also indicate clearly a lead author for this group along with a contact email address.

10. Please remove all personal information, ensure that the data shared are in accordance with participant consent, and re-upload a fully anonymized data set.

Additional guidance on preparing raw data for publication can be found in our Data Policy (https://journals.plos.org/plosone/s/data-availability#loc-human-research-participant-data-and-other-sensitive-data) and in the following article: http://www.bmj.com/content/340/bmj.c181.long .

Additional Editor Comments:

Both Reviewers and I are positive about this manuscript; it is well-written and advances our understanding of healthcare professionals’ knowledge and attitudes toward placebo and nocebo effects.

However, Reviewers raised comments that need to be addressed. So, I'd like to invite the authors to revise their manuscript and resubmit it for further evaluation properly.

Reviewers' comments:

Reviewer's Responses to Questions

**Comments to the Author**

1. Is the manuscript technically sound, and do the data support the conclusions?

Reviewer #1: Yes

Reviewer #2: Yes

2. Has the statistical analysis been performed appropriately and rigorously?

Reviewer #1: Yes

Reviewer #2: Yes

3. Have the authors made all data underlying the findings in their manuscript fully available?

Reviewer #1: No

Reviewer #2: Yes

4. Is the manuscript presented in an intelligible fashion and written in standard English?

Reviewer #1: Yes

Reviewer #2: Yes

Reviewer #1: Please see the attached document.

QUOTE FROM DOCUMENT:

SUMMARY OF MANUSCRIPT REVIEW

This manuscript is clearly written and demonstrates a rigorously executed mixed-method study that adds depth to our understanding of healthcare professionals’ knowledge and attitudes regarding placebo and nocebo effects. Particularly, I appreciated the correct use of descriptive statistics where incorrect use of p-values is widespread. While the study is promising and contributes valuable insights to the field, several minor revisions are needed to enhance clarity, accuracy, and consistency throughout the manuscript. I have also provided multiple suggestions for clarification and improvement. Particularly, the introduction would benefit from using more accurate references and deepening the rationale for the study’s contribution. The results and discussion contain several ambiguities or inconsistencies that need to be addressed. In the discussion, overgeneralization from qualitative findings should be avoided, and several limitations should be acknowledged due to the sample size and composition.

Reviewer #2: The current manuscript examined health care professionals’ knowledge on placebo and nocebo effects by means of a survey.

I enjoyed reading the manuscript. Please see below for some minor comments.

Introduction: clear introduction, written to the point.

- Since it is a short come of previous studies, could be it an example of ‘diverse range of healthcare professionals’ be given in the last paragraph of the introduction?

Method:

- Recruitment: is there a flow chain of number of participants were approached, responded etc. to present a response rate? Were reminders sent?

- It is clear that the order of questions was not randomized, but were the topics counterbalanced?

- How much time, on average, did participants need to complete the survey?

- Were there any control items? If so, how many, and how many participants failed the control questions? And was their data omitted?

Results:

- Could numbers be listed next to percentages?

- Could figure 2 be transformed to a table? It seems easier to read the information there. And perhaps add the numbers of discipline from Figure 1 as well.

- Very interesting results in Figure 3 – was it also explained why the listed disciplines were believed to be most prone to placebo and nocebo effects?

Results/Discussion

- The results are added together – also perhaps due to the smaller sample sizes e.g., in psychologists and nurses. However, although the placebo and nocebo effects are psychological effects, the treatment provided by the professionals are different. That is, a side effect due to medicine might be experienced differently (e.g., ascribed to the pill) than to another person (e.g., a psychologist). I think this difference in treatment type might also influence results and interpretations of placebo/nocebo. In psychology, it is not clear what nocebo effects are, since therapeutic side effects is an understudied topic. This is discussed later on in the discussion, but if professionals do not have the knowledge about nocebo effects int heir own field, it is perhaps hard to conclude there is a knowledge gap when there is no knowledge to begin with, since e.g., RCTs do not take nocebo effects into account.

Writing:

- At times passive writing (e.g., Research indicates that Y) – usually everything before ‘that’ can be deleted without losing the meaning of the sentence.

- Method: n should be italic.

- Numbers under ten should be written fully. Sentences starting with numbers should be written fully.

- Typo p. 20; comma after Table4,

**Do you want your identity to be public for this peer review?** For information about this choice, including consent withdrawal, please see our Privacy Policy

Reviewer #1: **Yes:** Leo Druart

Reviewer #2: **Yes:** Sanne Houben

---

## [Author Response · Author response to Decision Letter 1]

14 Oct 2025

Please see the Response to Reviewers and Cover letter uploaded. Thank you

---

## [Decision Letter · Decision Letter 1]

18 Nov 2025

Dear Dr. Bagnis,

Thank you for submitting your manuscript to PLOS ONE. After careful consideration, we feel that it has merit but does not fully meet PLOS ONE’s publication criteria as it currently stands. Therefore, we invite you to submit a revised version of the manuscript that addresses the points raised during the review process.

We look forward to receiving your revised manuscript.

Kind regards,

Vilfredo De Pascalis

Academic Editor

PLOS ONE

Journal Requirements:

Additional Editor Comments:

Reviewer 1 has accepted the revised manuscript for publication, while Reviewer 3 has suggested a minor revision. Thus, I invite the authors to address the two minor but essential comments raised by Reviewer 3 and to resubmit the revised manuscript, together with the response letter, for acceptance.

Reviewers' comments:

Reviewer's Responses to Questions

**Comments to the Author**

Reviewer #1: All comments have been addressed

Reviewer #3: (No Response)

2. Is the manuscript technically sound, and do the data support the conclusions?

Reviewer #1: Yes

Reviewer #3: Yes

3. Has the statistical analysis been performed appropriately and rigorously?

Reviewer #1: Yes

Reviewer #3: Yes

4. Have the authors made all data underlying the findings in their manuscript fully available?

Reviewer #1: (No Response)

Reviewer #3: Yes

5. Is the manuscript presented in an intelligible fashion and written in standard English?

Reviewer #1: Yes

Reviewer #3: Yes

Reviewer #1: Thank you for submitting a revised version of the manuscript and your diligence in responding to the comments. Lovely work.

Reviewer #3: Introductory note (for transparency) :

I am joining the review process at a later stage. My comments take into account the authors’ revisions and responses since the first round. While I might not have raised exactly the same initial points as the round-one reviewers, it would be inappropriate to proceed as if starting from scratch. I therefore focus on the manuscript in its revised form, acknowledging both the improvements made and the issues that, in my view, still warrant clarification or refinement.

Against this backdrop, I now turn to my comments.

1. The study’s external validity is constrained by a convenience sample that under-represents (or does not explicitly report) major European countries, which may overstate the breadth of coverage. Although this limitation is partially addressed in the revision and responses, concerns about generalizability persist. The recruitment via specific networks, combined with the limited presence of large countries (e.g., France, Germany), may lead readers to infer a pan-European picture that the data do not fully support. To what extent can the findings be extrapolated beyond the overrepresented settings? How many respondents came from the largest European countries, and how does this distribution shape the interpretability of the descriptive estimates? Are the conclusions clearly framed as arising from a multi-country convenience sample rather than a continent-wide survey?

Given this, the title may be misleading. I recommend tempering it, e.g., “A multi-country convenience survey of European healthcare professionals on placebo/nocebo.”

2. The revised version of the manuscript substantially improves transparency and methodological reporting. To further strengthen the paper, you might consider briefly expanding the Discussion on a few residual threats to internal validity: potential order/priming effects due to a non-randomized question sequence; the absence of attention/quality checks or completion-time thresholds; possible misclassification arising from self-reported professional status; the role of incentives and recruitment channels (e.g., EFIC-linked networks) in shaping completion patterns within the sample; the lack of psychometric evidence for item clusters or composite constructs; and the absence of sensitivity analyses contrasting complete-only vs complete+partial respondents for headline estimates. A short paragraph acknowledging these points—without changing the descriptive aims—would help readers calibrate the robustness and interpretability of the findings.

**Do you want your identity to be public for this peer review?** For information about this choice, including consent withdrawal, please see our Privacy Policy

Reviewer #1: **Yes:** Leo Druart

Reviewer #3: No

---

## [Author Response · Author response to Decision Letter 2]

27 Nov 2025

Please see file attached (Response to Reviewers R2)

---

## [Decision Letter · Decision Letter 2]

30 Nov 2025

Placebo and Nocebo in Clinical Practice: An Online Cross-sectional Survey of Healthcare Professionals from European Countries on Views, Practices and Training Needs

PONE-D-25-28687R2

Dear Dr. Bagnis,

We’re pleased to inform you that your manuscript has been judged scientifically suitable for publication and will be formally accepted for publication once it meets all outstanding technical requirements.

Kind regards,

Vilfredo De Pascalis

Academic Editor

PLOS ONE

Additional Editor Comments (optional):

Dear Authors, thank you for addressing all raised comments. I and the reviewers are all glad to accept your manuscript for publication.

Reviewers' comments:

Reviewer's Responses to Questions

**Comments to the Author**

Reviewer #3: All comments have been addressed

2. Is the manuscript technically sound, and do the data support the conclusions?

Reviewer #3: Yes

3. Has the statistical analysis been performed appropriately and rigorously?

Reviewer #3: Yes

4. Have the authors made all data underlying the findings in their manuscript fully available?

Reviewer #3: Yes

5. Is the manuscript presented in an intelligible fashion and written in standard English?

Reviewer #3: Yes

Reviewer #3: Thank you for taking my comments into consideration. Congratulations on this publication. !

**Do you want your identity to be public for this peer review?** For information about this choice, including consent withdrawal, please see our Privacy Policy

Reviewer #3: No

---

## [Editor Report · Acceptance letter]

PONE-D-25-28687R2

PLOS One

Dear Dr. Bagnis,

I'm pleased to inform you that your manuscript has been deemed suitable for publication in PLOS One. Congratulations! Your manuscript is now being handed over to our production team.

Kind regards,

on behalf of

Prof. Vilfredo De Pascalis

Academic Editor

PLOS One